# Decrease of Pro-Angiogenic Monocytes Predicts Clinical Response to Anti-Angiogenic Treatment in Patients with Metastatic Renal Cell Carcinoma

**DOI:** 10.3390/cells11010017

**Published:** 2021-12-22

**Authors:** Stephane Oudard, Nadine Benhamouda, Bernard Escudier, Patrice Ravel, Thi Tran, Emeline Levionnois, Sylvie Negrier, Philippe Barthelemy, Jean François Berdah, Marine Gross-Goupil, Cora N. Sternberg, Petri Bono, Camillo Porta, Ugo De Giorgi, Omi Parikh, Robert Hawkins, Martin Highley, Jochen Wilke, Thomas Decker, Corinne Tanchot, Alain Gey, Magali Terme, Eric Tartour

**Affiliations:** 1APHP, Hôpital Européen Georges Pompidou, INSERM U970, PARCC, Université de Paris, 75020 Paris, France; nadine.benhamouda@aphp.fr (N.B.); thi.tran@inserm.fr (T.T.); levionnois.emeline@gmail.com (E.L.); corinne.tanchot@inserm.fr (C.T.); alain.gey@aphp.fr (A.G.); magali.terme@inserm.fr (M.T.); 2APHP, Service de Cancérologie, Hôpital Européen Georges Pompidou, Université de Paris, 75908 Paris, France; 3Department of Medical Oncology, Institut Gustave Roussy, CEDEX, 94805 Villejuif, France; bernard.escudier@gustaveroussy.fr; 4Cancer Bioinformatics and Systems Biology, Institut de Recherche en Cancérologie de Montpellier, Campus Val d’Aurelle, Université Montpellier, CEDEX 5, 34298 Montpellier, France; patrice.ravel@umontpellier.fr; 5Centre Léon Bérard Lyon, University Lyon 1, 69008 Lyon, France; sylvie.negrier@lyon.unicancer.fr; 6Institut de Cancérologie Strasbourg Europe, Strasbourg University Hospital, 67200 Strasbourg, France; p.barthelemy@icans.eu; 7Medical Oncology Unit, Hôpital Privé Toulon-Hyères, Sainte-Marguerite, 83400 Hyeres, France; Jf.berdah@wanadoo.fr; 8Department of Medical Oncology, Centre Hospitalier Universitaire de Bordeaux, Bordeaux, 31000 Bordeaux, France; marine.gross-goupil@chu-bordeaux.fr; 9Englander Institute for Precision Medicine, Weill Cornell Medicine, Sandra and Edward Meyer Cancer, New York, NY 10065, USA; cns9006@med.cornell.edu; 10Kamppi Hospital Department, Terveystalo Finland, 00100 Helsinki, Finland; Petri.bono@terveystalo.com; 11Division of Translational Oncology, IRCCS San Matteo University Hospital, 27100 Pavia, Italy; camillo.porta@uniba.it; 12Division of Oncology, Policlinico Consorziale di Bari, University of Bari ‘A. Moro’, 70121 Bari, Italy; 13IRCCS Istituto Romagnolo per lo Studio dei Tumori (IRST) Dino Amadori, 47014 Meldola, Italy; ugo.degiorgi@irst.emr.it; 14Department of Oncology, Lancashire Teaching Hospitals NHS Foundation Trust, Preston PR2 9HT, UK; omi.parikh@lthtr.nhs.uk; 15Institute of Cancer Sciences, University of Manchester, Manchester M13 9PL, UK; Robert.e.hawkins@manchester.ac.uk; 16Oncology Centre, Derriford Hospital, Plymouth PL6 8DH, UK; martin.highley@nhs.net; 17Gemeinschaftspraxis Dres. Wilke/Wagner/Petzoldt, 90766 Fuerth, Germany; jw-studies@hotmail.de; 18Studienzentrum Onkologie, Practice for Hematology and Oncology, 88212 Ravensburg, Germany; Thomas.Decker@onkonet.eu

**Keywords:** angiogenesis, myeloid cells, biomarker, pro-angiogenic monocytes, renal cell carcinoma

## Abstract

The modulation of subpopulations of pro-angiogenic monocytes (VEGFR-1^+^CD14 and Tie2^+^CD14) was analyzed in an ancillary study from the prospective PazopanIb versus Sunitinib patient preferenCE Study (PISCES) (NCT01064310), where metastatic renal cell carcinoma (mRCC) patients were treated with two anti-angiogenic drugs, either sunitinib or pazopanib. Blood samples from 86 patients were collected prospectively at baseline (T1), and at 10 weeks (T2) and 20 weeks (T3) after starting anti-angiogenic therapy. Various subpopulations of myeloid cells (monocytes, VEGFR-1^+^CD14 and Tie2^+^CD14 cells) decreased during treatment. When patients were divided into two subgroups with a decrease (defined as a >20% reduction from baseline value) (group 1) or not (group 2) at T3 for VEGFR-1^+^CD14 cells, group 1 patients presented a median PFS and OS of 24 months and 37 months, respectively, compared with a median PFS of 9 months (*p* = 0.032) and a median OS of 16 months (*p* = 0.033) in group 2 patients. The reduction in Tie2^+^CD14 at T3 predicted a benefit in OS at 18 months after therapy (*p* = 0.04). In conclusion, in this prospective clinical trial, a significant decrease in subpopulations of pro-angiogenic monocytes was associated with clinical response to anti-angiogenic drugs in patients with mRCC.

## 1. Introduction

The influence of angiogenesis on different lymphocyte populations and on the phenotype and function of these populations has been well established. Inhibiting angiogenesis often results in reactivation of effector lymphocytes or inhibition of regulatory T cells (Treg) [1,2,3,4,5,6,7]. Myeloid cells, another component of immunity, are also essential for tumor angiogenesis which is a hallmark of cancer. A unique subpopulation of monocytes with pro-angiogenic, pro-lymphangiogenic, tissue remodeling and pro-tumor activity was described by De Palma et al. in 2005 [8]. They were identified by the Tie2/Tek receptor tyrosine kinase for angiopoietins 1–4 [9]. Tie2-expressing monocytes (TEM) are present in the blood and enriched in the intermediate subsets of monocytes (CD14^+^CD16^+^). This population expresses the macrophage colony-stimulating factor receptor (M-CSFR), colony-stimulating factor 1 receptor (CSF-1R) and C-C chemokine receptor type 5 (CCR5) markers but not vascular endothelial growth factor receptor-2 (VEGFR-2) or CCR2 and are close to M2 macrophages because they express arginase, CD163 and are more sensitive to T helper 2 (TH2) stimuli [10,11].

These TEMs are a more pro-angiogenic population than Tie2 negative monocytes. This characteristic can be explained by their higher expression of angiogenesis-related markers such as matrix metalloproteinase-9 (MMP-9), VEGFA, VEGFB, platelet-derived growth factor B (PDGFB), semaphorin 4D (SEMA-4D), lactadherin (Mfge8), cyclooxygenase-2 (COX-2), and WNT5A [9,11]. TEMs account for most of the proangiogenic activity of myeloid cells in tumors [8,12]. They are seen in some perivascular spaces in tumors, where they express high levels of basic fibroblast growth factor (bFGF) and insulin-like growth factor 1 (IGF1), which play a role in their pro-angiogenic and metastatic activities [8,13]. Genetic ablation of Tie2 on TEM has been shown to completely prevent tumor angiogenesis [8,14]. Expression of angiopoietin-2 by tumor cells induces the recruitment of Tie2-expressing monocytes in the tumor [15]. Hypoxia augmented the inhibitory effect of Ang-2 on the release of pro-inflammatory cytokines such as tumor necrosis factor α (TNFα) and the anti-angiogenic cytokine, interleukin (IL)-12 by monocytes [16]. It has also been reported that TEM infiltrating breast tumors endows lymphangiogenic and pro-metastatic activity [17,18].

TEM also exhibit immunosuppressive activity, as they decrease in vitro tumor-specific T cell proliferation mediated by dendritic cells and favor the conversion of CD4^+^T cells into Treg [18]. These immunosuppressive functions of TEM are mediated by Tie2 and VEGFR [18,19,20]. Angiopoietin 2 stimulates Tie2 monocytes to suppress T cell activation and promote regulatory T cell expansion via the upregulation of IL-10 and C-C motif chemokine ligand 17 (CCL17) [9], a chemokine favouring the recruitment of CCR4 positive Treg [21].

In both the murine models and primary isolated human cells, a subpopulation of monocytes/macrophages expressing VEGFR-1 has also been found to be highly angiogenic and to support metastasis growth [22]. It is known that Fms-related receptor tyrosine kinase 1 (Flt-1) signaling in myeloid cells is essential for the amplification of the angiogenic response and to promote glioma growth [23]. Moreover, in lung, pancreatic, and breast cancer models, Flt1 signaling in tumor-associated macrophage (TAM) has been shown to be necessary to support tumor angiogenesis [24]. Furthermore, there is a link between these two populations of TEMs and VEGFR-1 expressing monocytes because a subpopulation of TEMs express VEGFR-1 [9].

Renal cell carcinoma (RCC) is one of the most vascularized tumors due to the deletion, mutation, or promoter hypermethylation of the von Hippel–Lindau gene which is common (70%) in clear-cell RCC, accounting for 75% of RCCs [25,26]. These abnormalities lead to an upregulation of hypoxia-inducible factor (HIF) target genes, mainly those encoding for VEGF and angiopoietin-2, thus favoring angiogenesis [25,27]. This may explain why anti-angiogenic therapies have been the treatment of choice for kidney cancer as monotherapy and are now indicated in combination with anti-programmed death-1 (PD-1) immunotherapy [28,29,30].

Molecular classification of kidney cancer has revealed that certain prognostic groups rich in angiogenesis-related genes are associated with a better response to the anti-angiogenic drug sunitinib [31,32,33]. However, there are no blood biomarkers currently used in the management of patients treated with anti-angiogenic monotherapy or combination therapy.

We had the opportunity to analyze these pro-angiogenic monocyte populations in an ancillary study of the PazopanIb versus Sunitinib patient preferenCE Study (PISCES) which compared the quality of life in patients with metastatic RCC after first-line treatment with these two anti-angiogenic drugs which are tyrosine kinase inhibitors. PISCES was a prospective randomized, double-blind, multicenter, phase IIIb crossover study. Results showed a patient preference for pazopanib versus sunitinib [34], even if possible biases in certain methodological aspects prompted caution about drawing definitive conclusions [35]. A previous study, Comparz, showed comparable efficacy of these two tyrosine kinase inhibitors [36].

Our results show immunomodulation of these pro-angiogenic monocyte populations expressing Tie2 or VEGFR-1 in the course of antiangiogenic therapy. The decrease in the concentrations of these populations may be associated with clinical response to anti-angiogenic drugs.

## 2. Patients and Methods

### 2.1. Clinical Protocol

Samples from 67 patients with mRCC (22 female and 45 male) were included in the study. They comprised 60 clear cell RCC, 3 papillary adenocarcinoma, 2 chromophobe and 2 renal cancers of unknown histology. The Eastern Cooperative Oncology Group (ECOG) performance status was 0 in 53 patients and 1 in 14 patients. Based on Response Evaluation Criteria for Solid Tumors (RECIST) 1.1 criteria for best response, the results for this cohort were: 4 complete responses (CR), 27 partial responses (PR), 33 stable diseases (SD) and 2 progressive diseases (PD), while 1 patient was unevaluable for clinical response. This biomarker analysis was an ancillary study of the PISCES clinical trial. Metastatic RCC patients were randomly assigned to pazopanib 800 mg per day for 10 weeks, followed by a 2-week washout, then sunitinib 50 mg per day according to a 4 weeks on, 2 weeks off, 4 weeks on schedule (i.e., total of 10 weeks’ treatment) for 10 weeks, or the reverse sequence (Figure 1). Blood samples from 67 patients were collected prospectively at baseline (T1), then at 10 (T2) and 20 weeks (T3) after the start of therapy. The only reason for sampling at 20 weeks for T3 instead of at the end of the study at 22 weeks is that a fairly important biological assessment, particularly toxicological, was planned for this study at W22. All samples were cryopreserved and analyzed after thawing. Concentrations of total monocytes per liter and neutrophils were calculated using a full blood analyzer (LH750 Beckman-Coulter, Villepinte 95942 Roissy CDG, France). Pro-angiogenic monocytes were measured using multiparametric cytometry analysis.

All patients provided written informed consent prior to inclusion in the study. The protocol was reviewed and approved by the Ethical Committee/Institutional Review Board (IRAS ID 40179, EUDRACT 2009-014249-10, REC 10/H1010/11) and the study was conducted according to the Declaration of Helsinki and European Good Clinical Practice requirements. The study was registered with ClinicalTrials.gov (NCT01064310).

### 2.2. Flow Cytometry

Human cells (10^6^) diluted in Hanks’ balanced salt solution (HBSS) were incubated for 30 min at 4 °C with viability eFluor 450 dye (eBioscience-Fisher Scientific, 67403 Illkirch, France) and washed twice with the staining buffer (eBioscience) following the manufacturer’s protocol. After FcR blockage (Milteny Biotec, Paris, France), cells were stained with the respective antibody in HBSS 1% bovine serum albumin (BSA) for 30 min at 4 °C then washed twice before acquisition. Isotype controls were included in each experiment as well as live dead reagents to exclude dead cells (Dye e Fluor 450; eBioscience). Data acquisition was performed with a Navios flow cytometer (Beckman Coulter) and analyses were performed with Kaluza software (Beckman Coulter, Villepinte 95942 Roissy CDG, France) as previously described [37]. A detailed list of the antibodies used is shown in Table 1.

### 2.3. Mice Experiments

Eight-to-ten week-old female Balb/c mice were purchased from Janviers and kept in specific-pathogen-free conditions with filtertop cage at the INSERM U970 animal facility. Experimental protocols were approved by Paris Descartes University ethical committee (CEEA 34: agreement number 20931) in accordance with European guidelines (EC2010/63).

CT26, colon carcinoma and B16F10 melanoma cell lines were obtained from American Type Culture Collection and were cultured in RPMI 1640 (Life Technologies, 91941 Courtaboeuf, France) supplemented with 10% heat-inactivated fetal calf serum (GE Healthcare, 67403 Illkirch, France), 1 mM sodium pyruvate (Life Technologies, 91941 Courtaboeuf, France), 1 mM non-essential amino acids (Life Technologies, 91941 Courtaboeuf, France), 100 U/mL penicillin 100 mg/mL streptomycin (Life Technologies), 0.5 mM 2-b mercaptoethanol (Life Technologies, 91941 Courtaboeuf, France), and incubated at 37 °C in 5% CO_2_. We chose the 2 tumor models CT26 and B16F10 because in preliminary experiments we had shown that these tumors were infiltrated by pro-angiogenic monocytes.

CT26 cells (2 × 10^5^) were injected subcutaneously at day 0 in the right flank of Balb/c mice. When the tumors reached 9–10 mm^2^, the mice were treated with sunitinib by oral gavage at 40 mg/kg daily or DMSO 10% in phosphate-buffered saline (PBS). Tumor growth was monitored twice a week using a caliper. Mice were sacrificed at D20 post tumor graft. Immune cells were analyzed in blood and tumors by flow cytometry. Briefly, blood was collected and tumors were harvested and dissociated mechanically with MacsDissociator (Miltenyi Biotec, Paris, France). After FcR blocking with CD16/32 Ab (clone 93, eBioscience), the cells were stained with anti-mouse CD45 AF700 (clone30-F11, eBioscience), CD11b APCefluo780 (clone M1/70, eBioscience), Tie2 PE (clone TEK4, eBioscience), VEGFR-1 APC (Clone #141,522 R&D system, Abingdon OX14 3NB, United Kingdom) and the live/dead cell aqua blue viability (Life Technologies, 91941 Courtaboeuf, France). The antibodies were used according to manufacturers’ protocols. Acquisitions were performed on BD Fortessa X20 (Becton Dickinson, 38801 Le Pont de Claix France), and data were analyzed with FlowJo Software (BD).

## 3. Statistical Analyses

The log rank test was selected to compare groups for progression-free survival (PFS) and overall survival (OS) assessed by each investigator. Median follow up for survival was 34 months.

## 4. Results

### 4.1. Decrease of Different Myeloid Subpopulations after Anti-Angiogenic Treatment

Various subpopulations of myeloid cells (monocytes, VEGFR-1^+^CD14 and Tie2^+^CD14 cells) were defined by cytometry in the peripheral blood of RCC patients (Figure 2).

An association was observed between the two pro-angiogenic monocyte populations in our study. Thus, Tie2 was expressed by 11% of monocytes and VEGFR-1 by 9% of monocytes and 5% of monocytes co-expressed VEGFR-1 and Tie2 before treatment. In addition, Tie2 was expressed in 65% of the VEGFR-1^+^CD14 population and VEGFR-1 was expressed in 46% of the Tie2^+^CD14 population.

All of these populations decreased during treatment (Figure 3).

Thus, the mean concentrations of total monocytes decreased from 0.6 G/L at T1 to 0.4 G/L at T3 (*p* < 0.0001) (Figure 3A). The mean concentration of neutrophils also decreased from T1–T3 (Figure 3F) and the percentage of CD14^+^ among CD45^+^total cells was reduced from 21.6% at T1 to 13.13% at T3 (*p* < 0.0001) (Figure 3B). The mean percentage of VEGFR-1^+^CD14 among CD45^+^ cells decreased from 1.84% before therapy to 0.51% at the end of treatment (*p* < 0.0001: Figure 3C). Similarly, the mean percentage of Tie2^+^CD14 among CD45^+^ cells decreased from 2.19% at T1 to 0.86% at T3 (*p* < 0.0001: Figure 3D). The population co-expressing Tie2 and VEGFR-1 also decreased significantly after treatment (Figure 3E).

No difference was observed in the decrease of these parameters depending on the sequence of sunitinib or pazopanib administration.

Reductions in the three myeloid populations were observed as early as T2 after the first cycle of anti-angiogenic treatment and did not decrease further at T3 at the end of treatment (Figure 3A–D).

### 4.2. Correlations between the Decrease of Different Myeloid Subpopulations and Clinical Response

None of the various myeloid populations at baseline predicted clinical response. Overall, 71% of patients had a decrease in total monocytes at T3, 87% had a decrease in the Tie2^+^CD14 population and 82% a decrease in the VEGFR-1^+^CD14 population. Similar reductions were observed at T2. When patients were divided into two subgroups according to the presence or absence of a reduction in VEGFR-1^+^CD14 cells patients with a >20% decrease from baseline had a median PFS and OS of 24 months and 37 months, respectively, (Figure 4A,B), whereas those with <20% decrease from baseline had a median PFS of 9 months (*p* = 0.032) and a median OS of 16 months (*p* = 0.033)(Figure 4A,B).

These results were confirmed when the decrease in VEGFR-1^+^CD14 cells at T3 was considered as a continuous variable (*p* = 0.028 for PFS and *p* = 0.003 for OS).

Patients with a reduction in VEGFR-1^+^CD14 at T3 were also more likely to have responded to antiangiogenic therapy defined according to best clinical response using RECIST criteria (*p* = 0.035). A decrease in Tie2^+^CD14 at T3 also predicted an OS benefit at 18 months after therapy (*p* = 0.04) (Figure 5). These results were confirmed when the decrease in Tie2^+^CD14 cells at T2 was considered as a continuous variable (*p* = 0.029 for OS). However, the decrease of Tie2^+^CD14 between T3–T1 defined with a cutoff value of 20% did not correlate with PFS (data not shown). Interestingly, the decrease in total monocytes or CD14 positive cells, as well as neutrophils by flow cytometry, did not correlate with differences in PFS or OS (Figure 6A–D and Appendix A).

### 4.3. Modulation of Intratumoral Pro-Angiogenic Monocyte Levels in a Mouse Tumor Model

Next, we turned to a murine cancer model to further assess this decrease in pro-angiogenic monocytes after treatment with sunitinib or pazopanib. In contrast to human findings, Tie2 and VEGFR-1 myeloid populations did not decrease in the blood of mice grafted with CT26 colorectal tumors treated with sunitinib (Figure 7A,B). Conversely, the Tie2^+^CD11b^+^ population decreased in the tumor of these mice at the end of treatment (*p* = 0.055), and a trend was observed for the VEGFR-1^+^CD11b^+^ population (Figure 7C,D). The population of monocytes co-expressing VEGFR-1 and Tie2 decreased significantly in the tumor (*p* < 0.01). We also did not see a clear decrease of Tie2^+^CD14 in a melanoma tumor model (B16F10) (Appendix A).

## 5. Discussion

After anti-angiogenic therapy with sunitinib or pazopanib, populations of proangiogenic monocytes (VEGFR-1^+^CD14 and Tie2^+^CD14) decreased in patients with mRCC. To our knowledge, this is the first time that a decrease in the TEM population has been reported after anti-angiogenic therapy in mRCC. This decrease occurs early, is apparent from 10 weeks after the start of treatment and correlates with improved survival. Patients without a decrease in VEGFR-1^+^CD14 and Tie2^+^CD14 monocytes were strongly associated with treatment resistance, thus a therapeutic modification could be considered if further larger studies validated these results. Interestingly, this monocyte population also decreased intratumorally in a murine tumor model treated with sunitinib but not in blood.

In contrast to the frequency of Tie2^+^ monocytes of around 10% in blood, this population increased to more than 60% among intratumoral monocytes in renal tumors [38]. In line with these results, Venneri et al. analyzed the frequency of TEMs in cancer specimens and found that 55 and 70% of CD14^+^ monocytes were TEMs in colorectal and lung carcinomas, respectively, [12]. It has also been reported that the frequency of tumor infiltrated TEM in mRCC correlated with tumor grade, disease stage, positive lymph node status and the presence of distant metastases [38]. In non-small cell lung cancer (NSCLC), a higher percentage of TEMs in peripheral blood has been associated with poorer overall survival [39].

Interestingly, in metastatic breast cancer patients, however, the frequency of TEMs in peripheral blood remained unchanged by chemotherapy (paclitaxel) or anti-VEGF (bevacizumab) therapy [40] suggesting that inhibition of VEGF signaling is insufficient to modulate this population. In terms of regulation of TEMs, CSF-1, TGFβ in combination with VEGF and TNFα in combination with PlGF and angiopoietin have been shown to increase Tie2 expression on monocytes [20,41].

We also observed a decrease in the VEGFR-1^+^CD14 population in the blood of patients treated with sunitinib/pazopanib which correlated with a significant clinical benefit. This result is supported by another study that showed that in patients with neuroendocrine tumors, sunitinib treatment reduced the population of CD14^+^ monocytes expressing VEGFR-1 or CXCR4; however, the prognostic impact of this decrease was not reported [42]. Furthermore, in agreement with our results, in patients with recurrent glioblastoma treated with aflibercept, an anti-angiogenic molecule that binds to VEGF and PlGF, a decrease of VEGFR-1^+^ monocytes from baseline to 24 h was associated with improved objective response [43].

Low VEGFR-1 gene expression levels in metastatic colorectal cancer were predictive of prolonged OS after antiangiogenic therapy [44] and a high percentage of VEGFR-1^+^ cells in liver metastasis from colorectal cancer have been associated with worse patient outcomes [22]. Therefore, different studies converge to attribute a deleterious role to this pro-angiogenic monocyte population expressing VEGFR-1.

As with other groups [42,45,46], we also found a decrease in the monocyte population which was not associated with a significant impact on survival. This reinforces the interest in analyzing these pro-angiogenic monocyte subpopulations. Previous studies have reported a decrease in other myeloid populations such as myeloid-derived suppressive cells (MDSCs) with a more heterogeneous phenotype [47,48]. This decrease improved the immunosuppression condition of T cells, but did not appear to influence patient survival [45,49]. The relationship between this population of pro-angiogenic monocytes described in this work and the recently described population of immunosuppressive monocytes expressing cMet is not yet known [50].

Overall, these results suggest that these pro-angiogenic populations should be targeted more specifically for therapeutic purposes. Anti-angiopoietin-2, anti-VEGFR-1 antibodies or drugs inhibiting Tie2 receptor signaling are available [17,51,52,53]. In addition, most TEMs co-express the leukocyte marker CD52, which is the target of alemtuzumab, a monoclonal antibody approved by the Food and Drug Administration (FDA) for the treatment of chronic lymphocytic leukemia [17].

In mice, we found a tendency for these populations to decrease in tumors after sunitinib treatment but not in the blood. In this model, sunitinib could inhibit migration of these cells to the tumor but without impacting migration from the marrow to the blood. There are other differences that could explain these discordant results between humans and mice. For example, in mice, the CD11b marker defines all myeloid cells, whereas in humans we specifically analyzed pro-angiogenic monocyte populations (CD14). The human study was performed in hypervascularized kidney tumors. The hypervascularization of murine tumors was not analyzed. It is interesting to see an impact of sunitinib on the Tie2^+^myeloid cell population in mice because, in humans, intratumoral TEM frequency in NSCLC showed a positive correlation with microvascular density suggesting that the decrease in TEM may have an impact on tumor angiogenesis.

One of the limits of our work is that anti-angiogenic therapies are now only indicated as optional first-line therapy for patients with a favorable prognosis [54]. Thus, it would be important to evaluate the predictive value of these populations when combining immunotherapy and anti-angiogenic therapy, the current standard of care in mRCC. Preliminary results provide a rationale to analyze the modulation of these pro-angiogenic monocyte populations in these situations. High pretreatment serum angiopoietin-1 which favors intratumoral recruitment and activation of TEM was associated with shorter overall survival in CTLA-4 and PD-1 blockade-treated patients. These treatments also increased serum angiopoietin-2 in many patients early after treatment initiation, whereas ipilimumab plus bevacizumab treatment decreased their serum concentrations [55].

In conclusion, this work demonstrated a decrease in pro-angiogenic monocyte populations during sunitinib and pazopanib treatment in patients with metastatic kidney cancer. This modulation may have an impact on the clinical course of patients and may allow early identification of patients resistant to these treatments.

## Figures and Tables

**Figure 1 cells-11-00017-f001:**
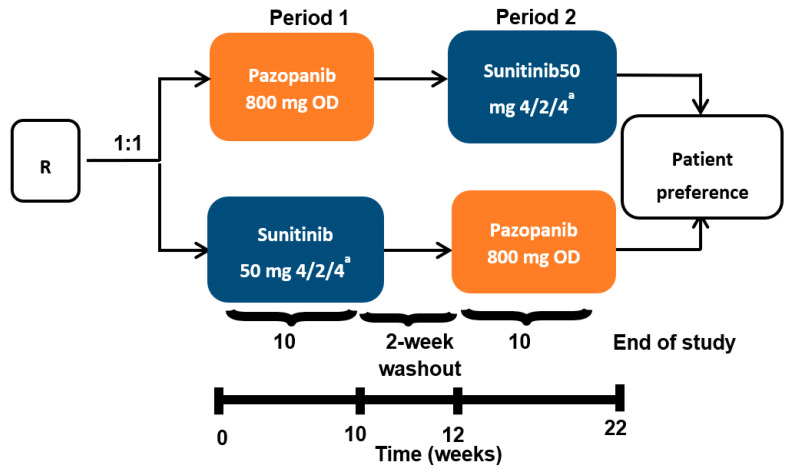
PISCES protocol and blood collection. **m**RCC patients were randomly assigned to pazopanib 800 mg per day for 10 weeks, followed by a 2-week washout, then sunitinib 50 mg per day (4 weeks on, 2 weeks off, 4 weeks on) for 10 weeks, or the reverse sequence. Blood samples from patients were collected prospectively at baseline (T1), 10 weeks (T2) and 20 weeks (T3) after the start of anti-angiogenic therapy. R stands for randomization. OD stands for oral administration once daily. ^a^ 4 weeks on treatment → 2 weeks of matching placebo → 4 weeks on treatment.

**Figure 2 cells-11-00017-f002:**
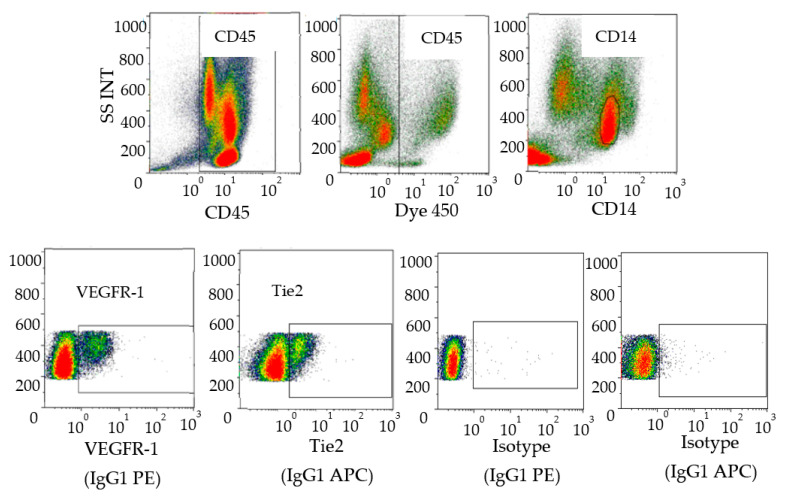
Characterization of human pro-angiogenic monocytes in blood. Peripheral blood mononuclear cells were stained with anti-CD45, anti-CD14, anti-VEGFR-1 and anti-Tie 2. The data analysis was performed as follows: Selection of CD45^+^ cells then live cells followed by gating on CD14^+^ cells and expression of VEGFR-1 or Tie-2^+^ cells on the monocyte population. Isotype controls corresponding to anti-VEGR-1 (IgG1 PE) and anti-Tie2 (IgG1 APC) wereshown (bottom right). Data acquisition was performed with a Navios (Beckman Coulter) and analyses were performed with Kaluza software (Beckman Coulter, Villepinte 95942 Roissy CDG, France).

**Figure 3 cells-11-00017-f003:**
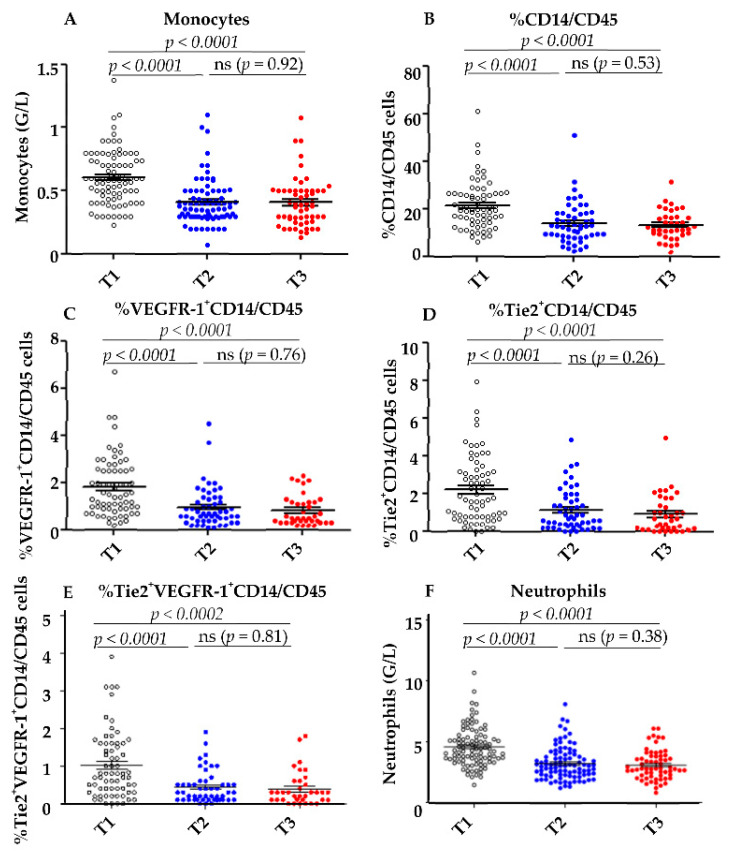
Decrease of myeloid cells during anti-angiogenic therapy. (**A**) The total number of monocytes and neutrophils (**F**) was determined by automatic hematologic cell count before (T1) and after the first cycle (T2) and at the end of anti-angiogenic therapy (T3). The percentage of CD14^+^ within CD45^+^ cells (**B**) and the percentage of VEGF-R1^+^ (**C**) and Tie2^+^ (**D**) and VEGFR-1^+^Tie2^+^ (**E**) within CD14^+^ cells were determined by cytometry before (T1), after the first cycle of anti-angiogenic therapy (T2), and at the end of anti-angiogenic therapy (T3). Statistical analysis was performed with GraphPad Prism^®^ software using the paired *t*-test.

**Figure 4 cells-11-00017-f004:**
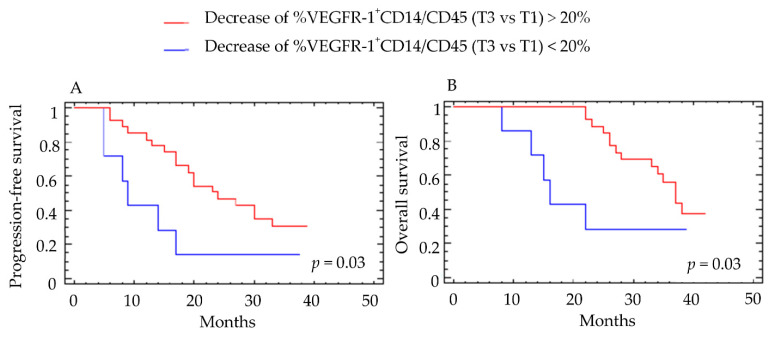
Correlation between the percent decrease in VEGFR-1^+^CD14/CD45^+^ cells and clinical outcome. The percentage of VEGFR-1^+^CD14/CD45^+^ cells was determined by cytometry before (T1) and at the end of anti-angiogenic therapy (T3). Patients were classified into two groups according to a cutoff value of 20% decrease in VEGFR-1^+^CD14/CD45^+^ cells. Results for (**A**) progression-free survival (PFS) and (**B**) overall survival (OS) are presented. For PFS, the group with >20% decrease included 40 patients and the group with <20% decrease included 12 patients. For OS, the group with >20% decrease included 45 patients and the group with <20% decrease included 13 patients. Log-rank test was used for the statistical analysis.

**Figure 5 cells-11-00017-f005:**
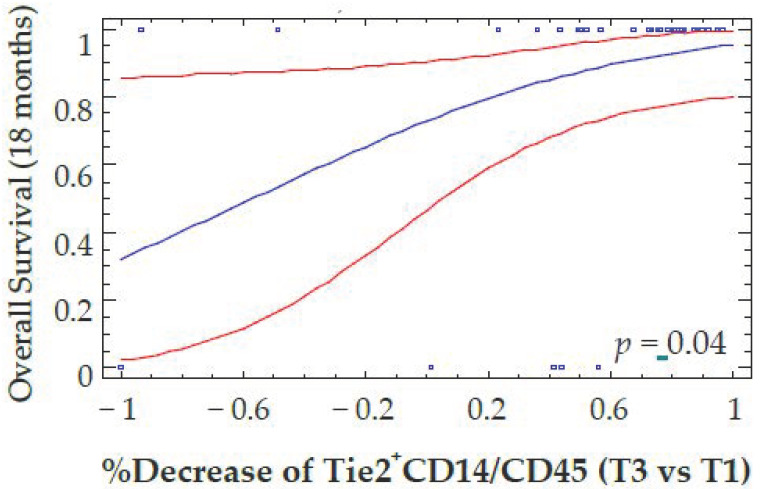
Correlation between the percent decrease of Tie2^+^CD14^/^CD45^+^ cells and overall Survival. The percentage of Tie2^+^CD14/CD45^+^ cells was determined by flow cytometry before (T1) and at the end of anti-angiogenic therapy (T3). The reduction in Tie2^+^CD14/CD45^+^ cells was plotted as a continuous variable of overall survival computed by a logistic regression model (blue line and its 95% confidence intervals red lines) with a time fixed at 18 months. Blue squares on the top indicate that corresponding patients are alive, while blue squares on the bottom correspond to patient death.

**Figure 6 cells-11-00017-f006:**
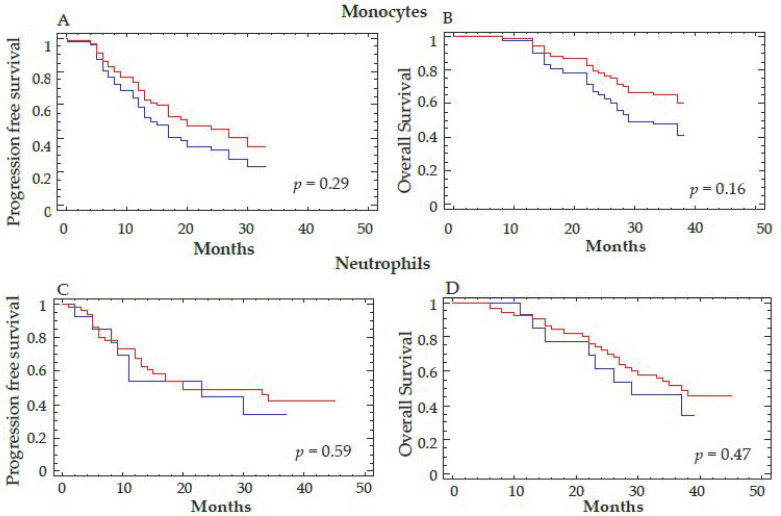
Correlation between the decrease in total monocytes and neutrophils and clinical outcome. The total number of monocytes (**A**,**B**) and neutrophils (**C**,**D**) were determined by automatic hematologic count before (T1) and at the end of anti-angiogenic therapy (T3). A cutoff of 20% for the percent decrease of monocytes and neutrophils was considered to separate the two groups of patients. Progression-free survival (**A**,**C**) and overall survival (**B**,**D**) are represented. Log-rank test was used for the statistical analysis. The red curve corresponds to a decrease in the total number of monocytes (**A**,**B**) or neutrophils (**C**,**D**) >20% and the blue curve to a decrease ≤20% for these 2 parameters between T3–T1.

**Figure 7 cells-11-00017-f007:**
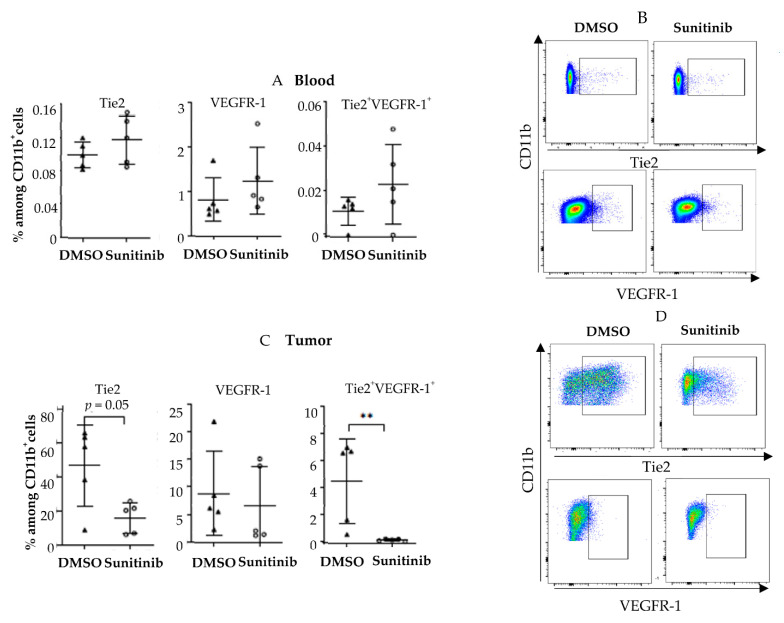
Sunitinib modulates pro-angiogenic monocytes in murine tumors. Balb/c mice were grafted subcutaneously with CT-26 tumor cells (0.2 × 10^5^ cells). When the tumors reached 9–10 mm^2^, mice were treated with sunitinib by oral gavage at 40 mg/kg daily or dimethylsulfoxide (DMSO) 10% in PBS. Mice were sacrificed at D20 post tumor graft. Analyses of Tie2^+^ VEGFR-1^+^ CD11b myeloid cells by flow cytometry in blood (**A**,**B**) and in tumors (**C**,**D**) are shown. Triangles and circles correspond to mice treated by DMSO and sunitinib respectively. Data are representative of two independent experiments with 5 mice/group. Data are mean ±SEM. Difference between the two groups were analyzed using with the Mann-whitney *t*-test. ** *p* < 0.01.

**Table 1 cells-11-00017-t001:** Detail of the antibodies used in the human study.

Antibody	Reference	Company
CD45KO	A96416	Beckman Coulter
CD14-APC H7	560180	Becton Dickinson
Tie2-APC	FAB3131A	R&D System
VEGFR-1-PE	FAB321P	R&D System
IgG1-APC	IC002A	R&D System
IgG1-PE	IC002P	R&D System

## Data Availability

The data presented in this study are available on request from the corresponding author. The data are not publicly available due to ethical consideration.

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
