# Peer review of "Decrease of Pro-Angiogenic Monocytes Predicts Clinical Response to Anti-Angiogenic Treatment in Patients with Metastatic Renal Cell Carcinoma"

_cells, 2021, doi:10.3390/cells11010017_

Round 1

Reviewer 1 Report

Summary
In this manuscript, authors investigate whether certain monocyte populations associated with angiogenesis can be used as a predictive biomarker for survival of renal cell carcinoma patients. I think the data from the PISCES trial are very clear – there is an association with decrease after starting therapy. However, the use of the murine models (1 colon cancer model is shown, authors mention a lung cancer model) did not make sense to me until the end of the discussion. Authors should really frame these experiments more clearly. For example. If authors  elaborate in the results section why they went to the murine models, and why they selected a colon cancer model and why/why not this is representative for the in vivo situation in RCC that would already help.

I really enjoyed reading the discussion, other sections of the manuscript were not written as clearly. If authors can significantly improve the way the manuscript is written, and better link the used mouse data, I think this manuscript could fit in Cells.

Comments and reccomendations

  1. Is it really necessary to add all email addresses of all authors? Having one or two corresponding authors should be sufficient and will result in a less cluttered title page.
  2. Please add what the trial name PISCES stands for in the abstract.
  3. Please consider thorough editing by an English (native) colleague or professional editor, as the manuscript sometimes suffers from too long sentences, typos or syntax errors. For instance, the first sentence of the introduction is quite long and should be split in 2, the second sentence has a space before a comma and “another” is one word, and a comma is missing after “immunity” in that sentence.
  4. The quality of the figures is really Please improve the figures so the data stands out better.
  5. Line 78: please add which other pro-inflammatory cytokine is also altered in release or rephrase sentence.
  6. Line 83: Treg is not previously defined (regulatory T cells are mentioned in the first sentence, I would suggest to add the abbreviation there).
  7. Line 116 – please elaborate a bit more on the monocyte populations, i.e. restate expressed markers and give a more defined conclusion like what is mentioned in the abstract.
  8. Line 121: remove . after 67 and use previously defined abbreviation of RCC. Furthermore, later on a breakdown between diseases is shown, where not all patients were clear cell patients; therefore, please remove from the first line of the patients and methods section.
  9. Line 130/figure 1
    1. Some of the text is illegible (mg is cut off). Please improve.
    2. What does superscript a stand or? Please add to figure legend.
    3. 4/2 should be 4/2/4, correct?
    4. What does OD stand for? Please add to figure legend.
    5. There is space for R to be written out in full (randomization) in the figure, otherwise put abbreviation at least in the figure legend.
    6. Pisces is written non-capitalized in the figure legend: please correct throughout to be consistent in the manuscript.
  10. Line 130 – were samples cryopreserved, or where all samples analyzed fresh? Please add this information. Furthermore, if cryopreserved samples were used, did authors check whether cryopreservation impact the # of monocytes in each population? As far as I am aware, not all cell populations, especially monocytes, do not like cryopreservation.
  11. Line 143 – please restate to Flow cytometry
  12. Line 144 – please add cell concentration used during staining, and the concentration of viability dye, or mention following manufacturer’s protocol.
  13. Line 148 – the live/dead information is double. Please remove.
  14. Line 149 – please add flow cytometer after Navios.
  15. Line 152 – please add antibody dilutions used.
  16. Line 154 – please add type of cage (i.e. filtertop).
  17. Line 169 – please add antibody dilutions used.
  18. Line 183 – half of the page is blank after this, please re-format.
  19. Line 183 – please use RCC consistently; adapt throughout the manuscript.
  20. Line 183/Figure 2 – very happy that researchers employed isotype controls; it would be great to add these panels to the figure to show specific gating.
  21. Line 191 – judging from figure 3, the percentage of cells expressing either marker is 2% (a bit less for VEGFR-1, a bit more for Tie2). Either the text here is wrong, or the axis on the graphs in figure 3.
  22. Line 195 –how was the concentration monocytes per liter calculated? As far as I am aware, this is difficult in flow cytometry and a full blood analyzer should be employed.
  23. Line 198 – indeed, here now the percentages observed in figure 3 are discussed. Where do the percentages in line 191 come from?
  24. Line 205 – please add this data so readers can evaluate this themselves, or remove this sentence. I would prefer adding this information as an extra figure/incorporate into figure 3.
  25. Line 220/Figure 3 – please also add a graph with the double expressing cells, and add the myeloid population/parameter also horizontal as graph title above each panel. Lastly, the significance indicators are not the same in all panels. Significance of P < 0001 does not make a lot of sense, do authors mean 0.0001? Please consider removing the non-significant values, or adding all of them for consistency throughout the graph.
  26. Line 207 – accentuate? Do authors mean decrease?
  27. Line 227/Figure 4 – please add the amount of patients in each group to the figure (legend).
  28. Line 238 – please change to flow cytometry
  29. Line 245/Figure 6 – please add the CD14 graphs, they fit just fine and would complete the picture.
  30. Line 251 – please add a linking sentence. For instance, “Next, we turned to a murine colon cancer model to further assess XXX”.
  31. Line 256 – “very” significantly us not very scientific language. Please remove very, and also re-phrase the sentence to reflect the right tense.
  32. Line 257 – either another cancer model, or in a lung tumor model. Furthermore, if this data exists, add as supplementary data to strengthen your message.
  33. Line 327 – another limitation is the used cancer model; is this really reflective of RCC as lung and colon cancer models were used.

Reviewer 2 Report

The manuscript, entitled “Decrease of Pro-angiogenic Monocytes Predicts Clinical Response to Anti-angiogenic Treatment in Patients With Metastatic Renal Cell Carcinoma”, demonstrates that a decrease in pro-angiogenic monocyte populations (VEGFR-1+CD14 and Tie2+CD14) during sunitinib and pazopanib treatment in metastatic kidney cancer patients from the prospective PISCES clinical trial 40 (NCT01064310). The significant decrease of subpopulations of pro-angiogenic monocytes, not total monocytes or CD14 positive cells, are associated with higher PFS or OS of patients, suggesting that the subpopulations of pro-angiogenic monocytes can be used as a blood marker for predicting the clinical outcome of patients treated with anti-angiogenic monotherapy or combination therapy.

This study reveals a clinical blood study from 86 mRCC patients and the results are solid and well organized. However, specific issues suggested for attention include:

Major issues:

  1. In the clinical protocol design, the blood samples from 86 patients were collected at baseline (T1), 10 weeks (T2), and 20 weeks (T3) after the start of anti-angiogenic therapy. Between the two anti-angiogenic drugs administration, there is a 2-week washout period, the end of the whole study should be 22 weeks, what is the rationale for setting T3 at 20 weeks instead of 22 weeks?
  2. In the mice experiments, the authors used CT26, a colon carcinoma cell line, to generate a mouse tumor model and detect pro-angiogenic monocyte levels in blood and tumor tissue.

1) As mentioned in the Discussion, since the human data was collected from RCC patients, why not use a renal carcinoma cell line to present this experiment?

2) “The lack of decrease of these populations in blood was confirmed in another lung tumor model (TC1) (results not shown)”. This data should be included in this manuscript.

  1. In order to reveal the specificity of the pro-angiogenic monocyte subpopulation that can be a blood marker for predicting the clinical outcome of patients after anti-angiogenic treatments, changes in other subpopulations of myeloid cells (e.g. granulocytes) should also be stated in this manuscript.
  2. Based on the decrease of Tie2+CD14+, how did the PFS change in patients after therapy?

Minor issue:

  1. Line 92, explain “TAMs”.
  2. Figure5, describe blue and 2 red lines in the figure.
  3. Figure6, describe blue and red lines in the figure.
  4. VEGFR-1+CD14, Tie2+CD14, VEGFR-1+CD14+, Tie2+CD14+, should be consistent.

Round 2

Reviewer 2 Report

 Responses to Reviewer 2

Major issues:

  1. In the clinical protocol design, the blood samples from 86 patients were collected at baseline (T1), 10 weeks (T2), and 20 weeks (T3) after the start of anti-angiogenic therapy. Between the two anti-angiogenic drugs administration, there is a 2-week washout period, the end of the whole study should be 22 weeks, what is the rationale for setting T3 at 20 weeks instead of 22 weeks?

We agree with the reviewer's comment. The only reason for sampling at 20 weeks for T3 instead of at the end of the study at 22 weeks is that a fairly important biological assessment, particularly toxicological, was planned for this study at W22. In order not to collect too much blood at the same time, this ancillary study on biomarkers used the samples taken at W18.

-Please give a brief description in the Methods based on the above explanation.

  1. In the mice experiments, the authors used CT26, a colon carcinoma cell line, to generate a mouse tumor model and detect pro-angiogenic monocyte levels in blood and tumor tissue.

1) As mentioned in the Discussion, since the human data was collected from RCC patients, why not use a renal carcinoma cell line to present this experiment?

We understand the reviewer's point. In fact there are very few mouse models of kidney tumors mimicking clear cell kidney tumors with VHL inactivation. Hu J's group developed such a model but tumor growth was very poor (Hu J et al Mol Ther Methods Clin Development 2018). We tested the Renca mouse tumor models (without VHL inactivation) and did not find significant infiltration of pro-angiogenic monocytes expressing Tie2. Therefore, we chose 2 other tumor models (CT26, B16F10) with a significant infiltration of pro-angiogenic monocytes expressing Tie2

-“we chose 2 other tumor models (CT26, B16F10) with a significant infiltration of pro-angiogenic monocytes expressing Tie2” add to the Methods.

2) “The lack of decrease of these populations in blood was confirmed in another lung tumor model (TC1) (results not shown)”. This data should be included in this manuscript.

Apart from the colon cancer model, CT26, we have added another mouse tumor model, B16F10 melanoma, where we observed no decrease in Tie2-expressing monocytes in blood but a decrease in tumor. However, we had only 2 mice in the control group , making statistical analysis difficult. This figure has been included as a supplementary figure 2.

-Since the data did not get a significant difference, it can not support the conclusion of “The lack of decrease of these populations in blood was confirmed in another lung tumor model (TC1)”. Please consider removing this sentence and supplemental data.

  1. In order to reveal the specificity of the pro-angiogenic monocyte subpopulation that can be a blood marker for predicting the clinical outcome of patients after anti-angiogenic treatments, changes in other subpopulations of myeloid cells (e.g. granulocytes) should also be stated in this manuscript.

According to your recommendations, we analyzed neutrophils during this treatment. We observed a decrease in neutrophil concentrations after treatment with anti-angiogenic drugs (Fig 3F). However, unlike pro-angiogenic monocytes, this decrease in neutrophils did not correlate with PFS or OS in these patients (Fig 6 CD). We have added these novel results in the manuscript (Page 7 : Line 301 and Page 8 : line 338).

-No additional concerns.

  1. Based on the decrease of Tie2+CD14+, how did the PFS change in patients after therapy?

When the parameter "decrease of Tie2+CD14+ between T3 and T1" is considered as a qualitative discrete variable cut-off at 20%, we did not find a statistically significant correlation with the PFS.

-Please add a brief description in that part.

Minor issue:

  1. Line 92, explain “TAMs”.

This abbreviation (tumor-associated macrophage) has been defined in the text.

  1. Figure5, describe blue and 2 red lines in the figure.

The legend of figure 5 has been revised and the blue and red lines better described

  1. Figure 6, describe blue and red lines in the figure.

The red curve corresponds to patients with a decrease in the total number of monocytes >20% and the red curve to those with a decrease ≤20% between T3 and T1. This detail has been added to the legend of the figure.

  1. VEGFR-1+CD14, Tie2+CD14, VEGFR-1+CD14+, Tie2+CD14+, should be consistent.

The wording of these various sub-populations has been rendered consistently throughout the manuscript.

-No additional concerns.

Author Response

Please find in the attached pdf file our answers to the reviewer 2

We have highlighted in blue our answers to the reviewer 2 questions highlighted in yellow to distinguish them from the 1st round review questions that we had already answered.
